# Global Filter Networks for Image Classification

**Yongming Rao**[*]  **Wenliang Zhao**[*]  **Zheng Zhu**  **Jiwen Lu**[†]  **Jie Zhou**

Department of Automation, Tsinghua University
State Key Lab of Intelligent Technologies and Systems
Beijing National Research Center for Information Science and Technology

## Abstract

Recent advances in self-attention and pure multi-layer perceptrons (MLP) models for vision have shown great potential in achieving promising performance with fewer inductive biases. These models are generally based on learning interaction among spatial locations from raw data. The complexity of self-attention and MLP grows quadratically as the image size increases, which makes these models hard to scale up when high-resolution features are required. In this paper, we present the Global Filter Network (GFNet), a conceptually simple yet computationally efficient architecture, that learns long-term spatial dependencies in the frequency domain with log-linear complexity. Our architecture replaces the self-attention layer in vision transformers with three key operations: a 2D discrete Fourier transform, an element-wise multiplication between frequency-domain features and learnable global filters, and a 2D inverse Fourier transform. We exhibit favorable accuracy/complexity trade-offs of our models on both ImageNet and downstream tasks. Our results demonstrate that GFNet can be a very competitive alternative to transformer-style models and CNNs in efficiency, generalization ability and robustness. Code is available at https://github.com/raoyongming/GFNet.

## 1 Introduction

The transformer architecture, originally designed for the natural language processing (NLP) tasks [42], has shown promising performance on various vision problems recently [10, 40, 27, 49, 4, 47, 35, 5]. Different from convolutional neural networks (CNNs), vision transformer models use self-attention layers to capture long-term dependencies, which are able to learn more diverse interactions between spatial locations. The pure multi-layer perceptrons (MLP) models [38, 39] further simplify the vision transformers by replacing the self-attention layers with MLPs that are applied across spatial locations. Since fewer inductive biases are introduced, these two kinds of models have the potential to learn more generic and flexible interactions among spatial locations from raw data.

One primary challenge of applying self-attention and pure MLP models to vision tasks is the considerable computational complexity that grows quadratically as the number of tokens increases. Therefore, typical vision transformer style models usually consider a relatively small resolution for the intermediate features (*e.g.* $14 \times 14$ tokens are extracted from the input images in both ViT [10] and MLP-Mixer [38]). This design may limit the applications of downstream dense prediction tasks like detection and segmentation. A possible solution is to replace the global self-attention with several local self-attention like Swin transformer [27]. Despite the effectiveness in practice, local self-attention brings quite a few hand-made choices (*e.g.*, window size, padding strategy, *etc*.) and limits the receptive field of each layer.

---

[*]Equal contribution.  [†]Corresponding author.

35th Conference on Neural Information Processing Systems (NeurIPS 2021).

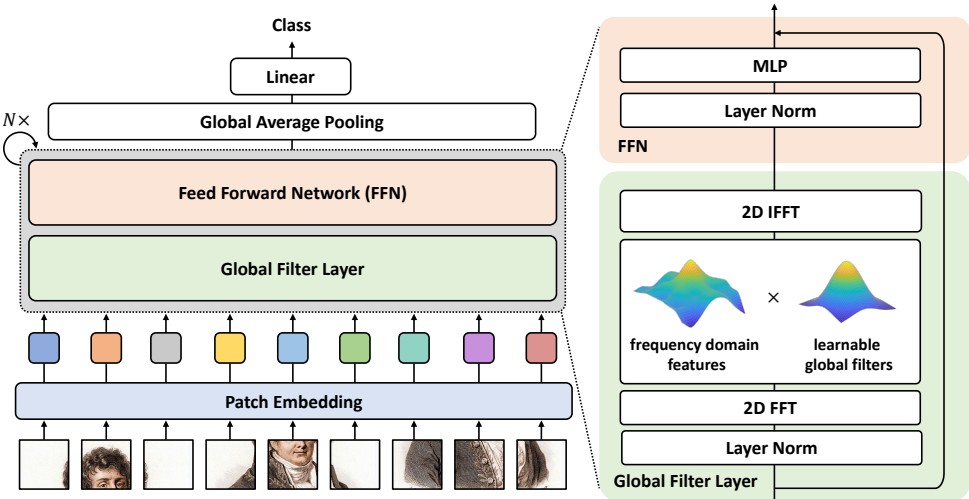

Figure 1: **The overall architecture of the Global Filter Network**. Our architecture is based on Vision Transformer (ViT) models with some minimal modifications. We replace the self-attention sub-layer with the proposed *global filter layer*, which consists of three key operations: a 2D discrete Fourier transform to convert the input spatial features to the frequency domain, an element-wise multiplication between frequency-domain features and the global filters, and a 2D inverse Fourier transform to map the features back to the spatial domain. The efficient fast Fourier transform (FFT) enables us to learn arbitrary interactions among spatial locations with log-linear complexity.

In this paper, we present a new conceptually simple yet computationally efficient architecture called Global Filter Network (*GFNet*), which follows the trend of removing inductive biases from vision models while enjoying the log-linear complexity in computation. The basic idea behind our architecture is to learn the interactions among spatial locations in the frequency domain. Different from the self-attention mechanism in vision transformers and the fully connected layers in MLP models, the interactions among tokens are modeled as a set of learnable *global filters* that are applied to the spectrum of the input features. Since the global filters are able to cover all the frequencies, our model can capture both long-term and short-term interactions. The filters are directly learned from the raw data without introducing human priors. Our architecture is largely based on the vision transformers only with some minimal modifications. We replace the self-attention sub-layer in vision transformers with three key operations: a 2D discrete Fourier transform to convert the input spatial features to the frequency domain, an element-wise multiplication between frequency-domain features and the global filters, and a 2D inverse Fourier transform to map the features back to the spatial domain. Since the Fourier transform is used to mix the information of different tokens, the global filter is much more efficient compared to the self-attention and MLP thanks to the $\mathcal{O}(L \log L)$ complexity of the fast Fourier transform algorithm (FFT) [7]. Benefiting from this, the proposed global filter layer is less sensitive to the token length $L$ and thus is compatible with larger feature maps and CNN-style hierarchical architectures *without modifications*. The overall architecture of GFNet is illustrated in Figure 1. We also compare our global filter with prevalent operations in deep vision models in Table 1.

Our experiments on ImageNet verify the effectiveness of GFNet. With a similar architecture, our model outperform the recent vision transformer and MLP models including DeiT [40], ResMLP [39] and gMLP [26]. When using the hierarchical architecture, GFNet can further enlarge the gap. GFNet also works well on downstream transfer learning and semantic segmentation tasks. Our results demonstrate that GFNet can be a very competitive alternative to transformer-style models and CNNs in efficiency, generalization ability and robustness.

## 2 Related works

**Vision transformers.** Since Dosovitskiy *et al.* [10] introduce transformers to the image classification and achieve a competitive performance compared to CNNs, transformers begin to exhibit their

| | Complexity (FLOPs) | # Parameters |
|---|---|---|
| Depthwise Convolution | $\mathcal{O}(k^2HWD)$ | $k^2D$ |
| Self-Attention | $\mathcal{O}(HWD^2 + H^2W^2D)$ | $4D^2$ |
| Spatial MLP | $\mathcal{O}(H^2W^2D)$ | $H^2W^2$ |
| **Global Filter** | $\mathcal{O}\left(HWD\lceil\log_2(HW)\rceil + HWD\right)$ | $HWD$ |

Table 1: Comparisons of the proposed *Global Filter* with prevalent operations in deep vision models. $H$, $W$ and $D$ are the height, width and the number of channels of the feature maps. $k$ is the kernel size of the convolution operation. The proposed global filter is much more efficient than self-attention and spatial MLP.

potential in various vision tasks [3, 4, 49]. Recently, there are a large number of works which aim to improve the transformers [40, 41, 27, 44, 18, 11, 48]. These works either seek for better training strategies [40, 11] or design better architectures [27, 44, 48] or both [41, 11]. However, most of the architecture modification of the transformers [44, 18, 27, 48] introduces additional inductive biases similar to CNNs. In this work, we only focus on the standard transformer architecture [10, 40] and our goal is to replace the heavy self-attention layer ($\mathcal{O}(L^2)$) to an more efficient operation which can still model the interactions among different spatial locations without introducing the inductive biases associated with CNNs.

**MLP-like models.** More recently, there are several works that question the importance of self-attention in the vision transformers and propose to use MLP to replace the self-attention layer in the transformers [38, 39, 26]. The MLP-Mixer [38] employs MLPs to perform token mixing and channel mixing alternatively in each block. ResMLP [39] adopts a similar idea but substitutes the Layer Normalization with an Affine transformation for acceleration. The recently proposed gMLP [26] uses a spatial gating unit to re-weight tokens in the spatial dimension. However, all of the above models include MLPs to mix the tokens spatially, which brings two drawbacks: (1) like the self-attention in the transformers, the spatial MLP still requires computational complexity quadratic to the length of tokens. (2) unlike transformers, MLP models are hard to scale up to higher resolution since the weights of the spatial MLPs have fixed sizes. Our work follows this trend and successfully resolves the above issues in MLP-like models. The proposed GFNet enjoys log-linear complexity and can be easily scaled up to any resolution.

**Applications of Fourier transform in vision.** Fourier transform has been an important tool in digital image processing for decades [32, 1]. With the breakthroughs of CNNs in vision [14, 13], there are a variety of works that start to incorporate Fourier transform in some deep learning method [24, 46, 9, 22, 6] for vision tasks. Some of these works employ discrete Fourier transform to convert the images to the frequency domain and leverage the frequency information to improve the performance in certain tasks [22, 46], while others utilize the convolution theorem to accelerate the CNNs via fast Fourier transform (FFT) [24, 9]. FFC [6] replaces the convolution in CNNs with an Local Fourier Unit and perform convolutions in the frequency domain. Very recent works also try to leverage Fourier transform to develop deep learning models to solve partial differential equations [25] and NLP tasks [23]. In this work, we propose to use learnable filters to interchange information globally among the tokens in the Fourier domain, inspired by the frequency filters in the digital image processing [32]. We also take advantage of some properties of FFT to reduce the computational costs and the number of parameters.

## 3 Method

### 3.1 Preliminaries: discrete Fourier transform

We start by introducing the discrete Fourier transform (DFT), which plays an important role in the area of digital signal processing and is a crucial component in our GFNet. For clarity, We first consider the 1D DFT. Given a sequence of $N$ complex numbers $x[n], 0 \leq n \leq N-1$, the 1D DFT converts the sequence into the frequency domain by:

$$X[k] = \sum_{n=0}^{N-1} x[n]e^{-j(2\pi/N)kn} := \sum_{n=0}^{N-1} x[n]W_N^{kn} \tag{3.1}$$

**Algorithm 1** Pseudocode of Global Filter Layer.

```
# x: the token features, B x H x W x D (where N = H * W)
# K: the frequency-domain filter, H x W_hat x D (where W_hat = W // 2 + 1, see Section 3.2 for details)

X = rfft2(x, dim=(1, 2))
X_tilde = X * K
x = irfft2(X_tilde, dim=(1, 2))
```

`rfft2/irfft2`: 2D FFT/IFFT for real signal

where $j$ is the imaginary unit and $W_N = e^{-j(2\pi/N)}$. The formulation of DFT in Equation (3.1) can be derived from the Fourier transform for continuous signal by sampling in both the time domain and the frequency domain (see Appendix A for details). Since $X[k]$ repeats on intervals of length $N$, it is suffice to take the value of $X[k]$ at $N$ consecutive points $k = 0, 1, \ldots, N-1$. Specifically, $X[k]$ represents to the spectrum of the sequence $x[n]$ at the frequency $\omega_k = 2\pi k/N$.

It is also worth noting that DFT is a one-to-one transformation. Given the DFT $X[k]$, we can recover the original signal $x[n]$ by the inverse DFT (IDFT):

$$x[n] = \frac{1}{N} \sum_{k=0}^{N-1} X[k] e^{j(2\pi/N)kn}. \tag{3.2}$$

For real input $x[n]$, it can be proved that (see Appendix A) its DFT is conjugate symmetric, i.e., $X[N-k] = X^*[k]$. The reverse is true as well: if we perform IDFT to $X[k]$ which is conjugate symmetric, a real discrete signal can be recovered. This property implies that the half of the DFT $\{X[k] : 0 \leq k \leq \lceil N/2 \rceil\}$ contains the full information about the frequency characteristics of $x[n]$.

DFT is widely used in modern signal processing algorithms for mainly two reasons: (1) the input and output of DFT are both discrete thus can be easily processed by computers; (2) there exist efficient algorithms for computing the DFT. The *fast Fourier transform* (FFT) algorithms take advantage of the symmetry and periodicity properties of $W_N^{kn}$ and reduce the complexity to compute DFT from $\mathcal{O}(N^2)$ to $\mathcal{O}(N \log N)$. The inverse DFT (3.2), which has a similar form to the DFT, can also be computed efficiently using the inverse fast Fourier transform (IFFT).

The DFT described above can be extend to 2D signals. Given the 2D signal $X[m, n], 0 \leq m \leq M-1, 0 \leq n \leq N-1$, the 2D DFT of $x[m, n]$ is given by:

$$X[u, v] = \sum_{m=0}^{M-1} \sum_{n=0}^{N-1} x[m, n] e^{-j2\pi\left(\frac{um}{M} + \frac{vn}{N}\right)}. \tag{3.3}$$

The 2D DFT can be viewed as performing 1D DFT on the two dimensions alternatively. Similar to 1D DFT, 2D DFT of real input $x[m, n]$ satisfied the conjugate symmetry property $X[M-u, N-v] = X^*[u, v]$. The FFT algorithms can also be applied to 2D DFT to improve computational efficiency.

## 3.2 Global Filter Networks

**Overall architecture.** Recent advances in vision transformers [10, 40] demonstrate that models based on self-attention can achieve competitive performance even without the inductive biases associated with the convolutions. Henceforth, there are several works [39, 38] that exploit approaches (*e.g.*, MLPs) other than self-attention to mix the information among the tokens. The proposed Global Filter Networks (GFNet) follows this line of work and aims to replace the heavy self-attention layer ($\mathcal{O}(N^2)$) with a simpler and more efficient one.

The overall architecture of our model is depicted in Figure 1. Our model takes as an input $H \times W$ non-overlapping patches and projects the flattened patches into $L = HW$ tokens with dimension $D$. The basic building block of GFNet consists of: 1) a *global filter layer* that can exchange spatial information efficiently ($\mathcal{O}(L \log L)$); 2) a feedforward network (FFN) as in [10, 40]. The output tokens of the last block are fed into a global average pooling layer followed by a linear classifier.

**Global filter layer.** We propose global filter layer as an alternative to the self-attention layer which can mix tokens representing different spatial locations. Given the tokens $x \in \mathbb{R}^{H \times W \times D}$, we first

perform 2D FFT (see Section 3.1) along the spatial dimensions to convert $\boldsymbol{x}$ to the frequency domain:

$$\boldsymbol{X} = \mathcal{F}[\boldsymbol{x}] \in \mathbb{C}^{H \times W \times D}, \tag{3.4}$$

where $\mathcal{F}[\cdot]$ denotes the 2D FFT. Note that $\boldsymbol{X}$ is a complex tensor and represents the spectrum of $\boldsymbol{x}$. We can then modulate the spectrum by multiplying a learnable filter $\boldsymbol{K} \in \mathbb{C}^{H \times W \times D}$ to the $\boldsymbol{X}$:

$$\tilde{\boldsymbol{X}} = \boldsymbol{K} \odot \boldsymbol{X}, \tag{3.5}$$

where $\odot$ is the element-wise multiplication (also known as the Hadamard product). The filter $\boldsymbol{K}$ is called the *global filter* since it has the same dimension with $\boldsymbol{X}$, which can represent an arbitrary filter in the frequency domain. Finally, we adopt the inverse FFT to transform the modulated spectrum $\tilde{X}$ back to the spatial domain and update the tokens:

$$\boldsymbol{x} \leftarrow \mathcal{F}^{-1}[\tilde{\boldsymbol{X}}]. \tag{3.6}$$

The formulation of the global filter layer is motivated by the frequency filters in the digital image processing [32], where the global filter $\boldsymbol{K}$ can be regarded as a set of learnable frequency filters for different hidden dimensions. It can be proved (see Appendix A) that the global filter layer is equivalent to a depthwise *global circular convolution* with the filter size $H \times W$. Therefore, the global filter layer is different from the standard convolutional layer which adopts a relatively small filter size to enforce the inductive biases of the locality. We also find although the proposed global filter can also be interpreted as a spatial domain operation, the filters learned in our networks exhibit more clear patterns in the frequency domain than the spatial domain, which indicates our models tend to capture relation in the frequency domain instead of spatial domain (see Figure 4). Note that the global filter implemented in the frequency domain is also much more efficient compared to the spatial domain, which enjoys a complexity of $\mathcal{O}(DL \log L)$ while the vanilla depthwise global circular convolution in the spatial domain has $\mathcal{O}(DL^2)$ complexity. We will also show that the global filter layer is better than its local convolution counterparts in the experiments.

It is also worth noting that in the implementation, we make use of the property of DFT to reduce the redundant computation. Since $\boldsymbol{x}$ is a real tensor, its DFT $\boldsymbol{X}$ is conjugate symmetric, *i.e.* $\boldsymbol{X}[H - u, W - v, :] = \boldsymbol{X}^*[H, W, :]$. Therefore, we can take only the half of the values in the $\boldsymbol{X}$ but preserve the full information at the same time:

$$\boldsymbol{X}_r = \boldsymbol{X}[:, 0 : \widehat{W}] := \mathcal{F}_r[\boldsymbol{x}], \quad \widehat{W} = \lceil W/2 \rceil, \tag{3.7}$$

where $\mathcal{F}_r$ denotes the 2D FFT for real input. In this way, we can implement the global filter as $\boldsymbol{K}_r \in \mathbb{C}^{H \times \widehat{W} \times D}$, which can reduce half the parameters. This can also ensure $\mathcal{F}_r^{-1}[\boldsymbol{K}_r \odot \boldsymbol{X}_r]$ is a real tensor, thus it can be added directly to the input $\boldsymbol{x}$. The global filter layer can be easily in modern deep learning frameworks (*e.g.*, PyTorch [31]), as is shown in Algorithm 1. The FFT and IFFT are well supported by GPU and CPU thanks to the acceleration libraries like `cuFFT` and `mkl-fft`, which makes our models perform well on hardware.

**Relationship to other transformer-style models.** The GFNet follows the line of research about the exploration of approaches to mix the tokens. Compared to existing architectures like vision transformers and pure MLP models, we exhibit that GFNet has several favorable properties: 1) GFNet is more efficient. The complexity of both the vision transformers [10, 40, 41] and the MLP models [38, 39] is $\mathcal{O}(L^2)$. Different from them, global filter layer only consists an FFT ($\mathcal{O}(L \log L)$), an element-wise multiplication ($\mathcal{O}(L)$) and an IFFT ($\mathcal{O}(L \log L)$), which means the total computational complexity is $\mathcal{O}(L \log L)$. 2) Although pure MLP models are simpler compared to transformers, it is hard to fine-tune them on higher resolution (*e.g.*, from $224 \times 224$ resolution to $384 \times 384$ resolution) since they can only process a fixed number of tokens. As opposed to pure MLP models, we will show that our GFNet can be easily scaled up to higher resolution. Our model is more flexible since both the FFT and the IFFT have no learnable parameters and can process sequences with arbitrary length. We can simply interpolate the global filter $\boldsymbol{K}$ to $\boldsymbol{K}' \in \mathbb{C}^{H' \times W' \times D}$ for different inputs, where $H' \times W'$ is the target size. The interpolation is reasonable due to the property of DFT. Each element of the global filter $\boldsymbol{K}[u, v]$ corresponds to the spectrum of the filter at $\omega_u = 2\pi u/H, \omega_v = 2\pi v/W$ and thus, the global filter $\boldsymbol{K}$ can be viewed as a sampling of a continuous spectrum $\boldsymbol{K}(\omega_u, \omega_v)$, where $\omega_u, \omega_v \in [0, 2\pi]$. Hence, changing the resolution is equivalent to changing the sampling interval of $\boldsymbol{K}(\omega_u, \omega_v)$. Therefore, we only need to perform interpolation to shift from one resolution to another.

Table 2: Detailed configurations of different variants of GFNet. For hierarchical models, we provide the number of channels and blocks in 4 stages. The FLOPs are calculated with $224 \times 224$ input.

| Model | #Blocks | #Channels | Params (M) | FLOPs (G) |
|---|---|---|---|---|
| GFNet-Ti | 12 | 256 | 7 | 1.3 |
| GFNet-XS | 12 | 384 | 16 | 2.9 |
| GFNet-S | 19 | 384 | 25 | 4.5 |
| GFNet-B | 19 | 512 | 43 | 7.9 |
| GFNet-H-Ti | [3, 3, 10, 3] | [64, 128, 256, 512] | 15 | 2.1 |
| GFNet-H-S | [3, 3, 10, 3] | [96, 192, 384, 768] | 32 | 4.6 |
| GFNet-H-B | [3, 3, 27, 3] | [96, 192, 384, 768] | 54 | 8.6 |

We also notice recently a concurrent work FNet [23] leverages Fourier transform to mix tokens. Our work is distinct from FNet in three aspects: (1) FNet performs FFT to the input and directly adds the real part of the spectrum to the input tokens, which blends the information from different domains (spatial/frequency) together. On the other hand, GFNet draws motivation from the frequency filters, which is more reasonable. (2) FNet only keeps the real part of the spectrum. Note that the spectrum of real input is conjugate symmetric, which means the real part is exactly symmetric and thus contains redundant information. Our GFNet, however, utilizes this property to simplify the computation. (3) FNet is designed for NLP tasks, while our GFNet focuses on vision tasks. In our experiments, we also implement the FNet and show that our model outperforms it.

**Architecture variants.** Due to the limitation from the quadratic complexity in the self-attention, vision transformers [10, 40] are usually designed to process a relatively small feature map (*e.g.*, $14 \times 14$). However, our GFNet, which enjoys log-linear complexity, avoids that problem. Since in our GFNet the computational costs do not grow such significantly when the feature map size increases, we can adopt a hierarchical architecture inspired by the success of CNNs [21, 14]. Generally speaking, we can start from a large feature map (*e.g.*, $56 \times 56$) and gradually perform downsampling after a few blocks. In this paper, we mainly investigate two kinds of variants of GFNet: transformer-style models with a fixed number of tokens in each block and CNN-style hierarchical models with gradually downsampled tokens. For transformer-style models, we begin with a 12-layer model (*GFNet-XS*) with a similar architecture with DeiT-S and ResMLP-12. Then, we obtain 3 variants of the model (*GFNet-Ti*, *GFNet-S* and *GFNet-B*) by simply adjusting the depth and embedding dimension, which have similar computational costs with ResNet-18, 50 and 101 [14]. For hierarchical models, we also design three models (*GFNet-H-Ti*, *GFNet-H-S* and *GFNet-H-B*) that have these three levels of complexity following the design of PVT [43]. We use $4 \times 4$ patch embedding to form the input tokens and use a non-overlapping convolution layer to downsample tokens following [43, 27]. Unlike PVT [43] and Swin [27], we directly apply our building block on different stages without any modifications. The detailed architectures are summarized in Table 2.

## 4 Experiments

We conduct extensive experiments to verify the effectiveness of our GFNet. We present the main results on ImageNet [8] and compare them with various architectures. We also test our models on the downstream transfer learning datasets including CIFAR-10/100 [20], Stanford Cars [19] and Flowers-102 [30]. Lastly, we investigate the efficiency and robustness of the proposed models and provide visualization to have an intuitive understanding of our method.

### 4.1 ImageNet Classification

**Setups.** We conduct our main experiments on ImageNet [8], which is a widely used large-scale benchmark for image classification. ImageNet contains roughly 1.2M images from 1,000 categories. Following common practice [14, 40], we train our models on the training set of ImageNet and report the single-crop top-1 accuracy on 50,000 validation images. To fairly compare with previous works [40, 39], we follow the most training details for our models and do not add extra regularization methods like [18]. Different from [40], we does not use EMA model [33], RandomEarse [50] and

Table 3: **Comparisons with transformer-style architectures on ImageNet.** We compare different transformer-style architectures for image classification including vision transformers [40], MLP-like models [39, 26] and our models that have comparable FLOPs and the number of parameters. We report the top-1 accuracy on the validation set of ImageNet as well as the number of parameters and FLOPs. All of our models are trained with $224 \times 224$ images. We use "↑384" to represent models finetuned on $384 \times 384$ images for 30 epochs.

| Model | Params (M) | FLOPs (G) | Resolution | Top-1 Acc. (%) | Top-5 Acc. (%) |
|---|---|---|---|---|---|
| DeiT-Ti [40] | 5 | 1.2 | 224 | 72.2 | 91.1 |
| gMLP-Ti [26] | 6 | 1.4 | 224 | 72.0 | - |
| GFNet-Ti | 7 | 1.3 | 224 | 74.6 | 92.2 |
| ResMLP-12 [39] | 15 | 3.0 | 224 | 76.6 | - |
| GFNet-XS | 16 | 2.9 | 224 | 78.6 | 94.2 |
| DeiT-S [40] | 22 | 4.6 | 224 | 79.8 | 95.0 |
| gMLP-S [26] | 20 | 4.5 | 224 | 79.4 | - |
| GFNet-S | 25 | 4.5 | 224 | 80.0 | 94.9 |
| ResMLP-36 [39] | 45 | 8.9 | 224 | 79.7 | - |
| GFNet-B | 43 | 7.9 | 224 | 80.7 | 95.1 |
| GFNet-XS↑384 | 18 | 8.4 | 384 | 80.6 | 95.4 |
| DeiT-B [40] | 86 | 17.5 | 224 | 81.8 | 95.6 |
| gMLP-B [26] | 73 | 15.8 | 224 | 81.6 | - |
| GFNet-S↑384 | 28 | 13.2 | 384 | 81.7 | 95.8 |
| GFNet-B↑384 | 47 | 23.3 | 384 | 82.1 | 95.8 |

Table 4: **Comparisons with hierarchical architectures on ImageNet.** We compare different hierarchical architectures for image classification including convolutional neural networks [14, 34], hierarchical vision transformers [43, 27] and our hierarchical models that have comparable FLOPs and number of parameters. We report the top-1 accuracy on the validation set of ImageNet as well as the number of parameters and FLOPs. All models are trained and tested with $224 \times 224$ images.

| Model | Params (M) | FLOPs (G) | Top-1 Acc. (%) | Top-5 Acc. (%) |
|---|---|---|---|---|
| ResNet-18 [14] | 12 | 1.8 | 69.8 | 89.1 |
| RegNetY-1.6GF [34] | 11 | 1.6 | 78.0 | - |
| PVT-Ti [26] | 13 | 1.9 | 75.1 | - |
| GFNet-H-Ti | 15 | 2.1 | 80.1 | 95.1 |
| ResNet-50 [40] | 26 | 4.1 | 76.1 | 92.9 |
| RegNetY-4.0GF [34] | 21 | 4.0 | 80.0 | - |
| PVT-S [26] | 25 | 3.8 | 79.8 | - |
| Swin-Ti [27] | 29 | 4.5 | 81.3 | - |
| GFNet-H-S | 32 | 4.6 | 81.5 | 95.6 |
| ResNet-101 [40] | 45 | 7.9 | 77.4 | 93.5 |
| RegNetY-8.0GF [34] | 39 | 8.0 | 81.7 | - |
| PVT-M [26] | 44 | 6.7 | 81.2 | - |
| Swin-S [27] | 50 | 8.7 | 83.0 | - |
| GFNet-H-B | 54 | 8.6 | 82.9 | 96.2 |

repeated augmentation [17], which are important to train DeiT while sightly hurting the performance of our models. We set the gradient clipping norm to 1 for all of our models. During finetuning at the higher resolution, we use the hyper-parameters suggested by the implementation of [40] and train the model for 30 epochs. All of our models are trained on a single machine with 8 GPUs. More details can be found in Appendix B.

**Comparisons with transformer-style architectures.** The results are presented in Table 3. We compare our method with different transformer-style architectures for image classification including

vision transformers (DeiT [40]) and MLP-like models (ResMLP [39] and gMLP [26]) that have similar complexity and number of parameters. We see that our method can clearly outperform recent MLP-like models such as ResMLP [39] and gMLP [26], and show similar performance to DeiT. Specifically, GFNet-XS outperforms ResMLP-12 by 2.0% while having slightly fewer FLOPs. GFNet-S also achieves better top-1 accuracy compared to gMLP-S and DeiT-S. Our tiny model is significantly better compared to both DeiT-Ti (+2.4%) and gMLP-Ti (+2.6%) with the similar level of complexity.

**Comparisons with hierarchical architectures.** We compare different kinds of hierarchical models in Figure 4. ResNet [14] is the most widely used convolutional model while RegNet [34] is a family of carefully designed CNN models. We also compare with recent hierarchical vision transformers PVT [43] and Swin [27]. Benefiting from the log-linear complexity, GFNet-H models show significantly better performance than ResNet, RegNet and PVT and achieve similar performance with Swin while having a much simpler and more generic design.

**Fine-tuning at higher resolution.** One prominent problem of MLP-like models is that the feature resolution is not adjustable. On the contrary, the proposed global filter is more flexible. We demonstrate the advantage of GFNet by finetuning the model trained at $224 \times 224$ resolution to higher resolution following the practice in vision transformers [40]. As shown in Table 3, our model can easily adapt to higher resolution with only 30 epoch finetuning and achieve better performance.

Table 5: **Results on transfer learning datasets**. We report the top-1 accuracy on the four datasets as well as the number of parameters and FLOPs.

| Model | FLOPs | Params | CIFAR-10 | CIFAR-100 | Flowers-102 | Cars-196 |
|---|---|---|---|---|---|---|
| ResNet50 [14] | 4.1G | 26M | - | - | 96.2 | 90.0 |
| EfficientNet-B7 [37] | 37G | 66M | 98.9 | 91.7 | 98.8 | 94.7 |
| ViT-B/16 [10] | 55.4G | 86M | 98.1 | 87.1 | 89.5 | - |
| ViT-L/16 [10] | 190.7G | 307M | 97.9 | 86.4 | 89.7 | - |
| Deit-B/16 [40] | 17.5G | 86M | 99.1 | 90.8 | 98.4 | 92.1 |
| ResMLP-12 [39] | 3.0G | 15M | 98.1 | 87.0 | 97.4 | 84.6 |
| ResMLP-24 [39] | 6.0G | 30M | 98.7 | 89.5 | 97.9 | 89.5 |
| GFNet-XS | 2.9G | 16M | 98.6 | 89.1 | 98.1 | 92.8 |
| GFNet-H-B | 8.6G | 54M | 99.0 | 90.3 | 98.8 | 93.2 |

## 4.2 Transfer learning

To test the generality of our architecture and the learned representation, we evaluate GFNet on a set of commonly used transfer learning benchmark datasets including CIFAR-10 [20], CIFAR-100 [20], Stanford Cars [19] and Flowers-102 [30]. We follow the setting of previous works [37, 10, 40, 39], where the model is initialized by the ImageNet pre-trained weights and finetuned on the new datasets. We evaluate the transfer learning performance of our basic model and best model. The results are presented in Table 5. The proposed models generally work well on downstream datasets. GFNet models outperform ResMLP models by a large margin and achieve very competitive performance with state-of-the-art EfficientNet-B7. Our models also show competitive performance compared to state-of-the-art CNNs and vision transformers.

## 4.3 Analysis and visualization

**Efficiency of GFNet.** We demonstrate the efficiency of our GFNet in Figure 2, where the models are compared in theoretical FLOPs, actual latency and peak memory usage on GPU. We test a single building block of each model (including one token mixing layer and one FFN) with respect to the different numbers of tokens and set the feature dimension and batch size to 384 and 32 respectively. The self-attention model quickly runs out of memory when feature resolution exceeds $56^2$, which is also the feature resolution of our hierarchical model. The advantage of the proposed architecture becomes larger as the resolution increases, which strongly shows the potential of our model in vision tasks requiring high-resolution feature maps.

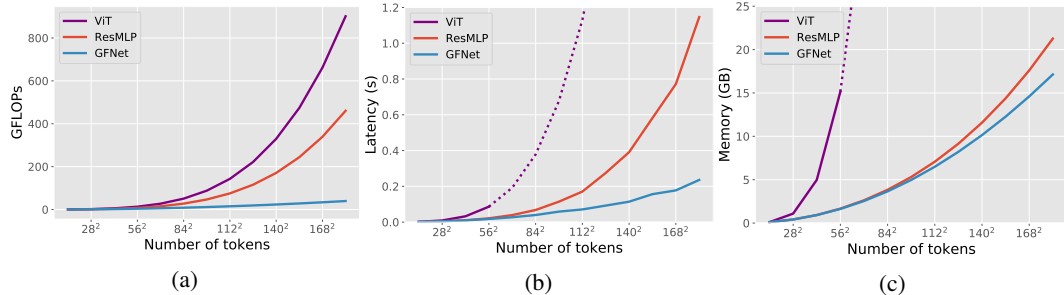

(a)                      (b)                      (c)

Figure 2: Comparisons among GFNet, ViT [10] and ResMLP [39] in **(a)** FLOPs **(b)** latency and **(c)** GPU memory with respect to the number of tokens (feature resolution). The dotted lines indicate the estimated values when the GPU memory has run out. The latency and GPU memory is measured using a single NVIDIA RTX 3090 GPU with batch size 32 and feature dimension 384.

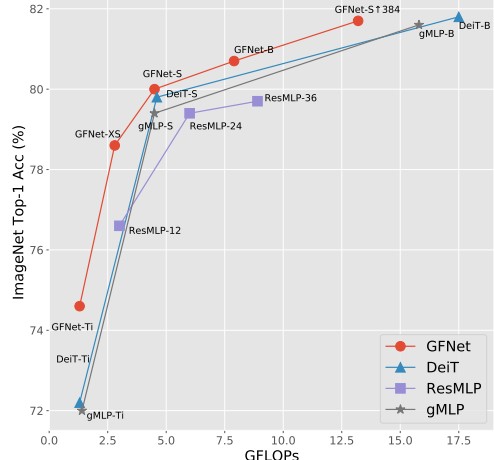

Figure 3: ImageNet acc. *vs* model complexity.

Table 6: Comparisons among the GFNet and other variants based on the transformer-like architecture on ImageNet. We show that GFNet outperforms the ResMLP [39], FNet [23] and models with local depth-wise convolutions. We also report the number of parameters and theoretical complexity in FLOPs.

| Model | Acc (%) | Param (M) | FLOPs (G) |
|---|---|---|---|
| DeiT-S [40] | 79.8 | 22 | 4.6 |
| Local Conv ($3 \times 3$) | 77.7 | 15 | 2.8 |
| Local Conv ($5 \times 5$) | 78.1 | 15 | 2.9 |
| Local Conv ($7 \times 7$) | 78.2 | 15 | 2.9 |
| ResMLP [39] | 76.6 | 15 | 3.0 |
| FNet [23] | 71.2 | 15 | 2.9 |
| GFNet-XS | 78.6 | 16 | 2.9 |

**Complexity/accuracy trade-offs.** We show the computational complexity and accuracy trade-offs of various transformer-style architectures in Figure 3. It is clear that GFNet achieves the best trade-off among all kinds of models.

**Ablation study on the global filter.** To more clearly show the effectiveness of the proposed global filters, we compare GFNet-XS with several baseline models that are equipped with different token mixing operations. The results are presented in Table 6. All models have a similar building block ( token mixing layer + FFN ) and the same feature dimension of $D = 384$. We also implement the recent FNet [23] for comparison, where a 1D FFT on feature dimension and a 2D FFT on spatial dimensions are used to mix tokens. As shown in Table 6, our method outperforms all baseline methods except DeiT-S that has 64% higher FLOPs.

**Robustness & generalization ability.** Inspired by the [29], we further conduct experiments to evaluate the robustness and the generalization ability of the GFNet. For robustness, we consider ImageNet-A, ImageNet-C, FGSM and PGD. ImageNet-A [16] (IN-A) is a challenging dataset that contains natural adversarial examples. ImageNet-C [15] (IN-C) is used to validate the robustness of the model under various types of corruption. We use the mean corruption error (mCE, lower is better) on ImageNet-C as the evaluation metric. FGSM [12] and PGD [28] are two widely used algorithms that are targeted to evaluate the adversarial robustness of the model by single-step attack and multi-step attack, respectively. For generalization ability, we adopt two variants of ImageNet validation set: ImageNet-V2 [36] (IN-V2) and ImageNet-Real [2] (IN-Real). ImageNet-V2 is a re-collected version of ImageNet validation set following the same data collection procedure of ImageNet, while ImageNet-Real contains the same images as ImageNet validation set but has reassessed labels. We compare GFNet-S with various baselines in Table 7 including CNNs, Transformers and MLP-like architectures and find the GFNet enjoys both favorable robustness and generalization ability.

Table 7: **Evaluation of robustness and generalization ability**. We measure the robustness from different aspects, including the adversarial robustness by adopting adversarial attack algorithms including FGSM and PGD and the performance on corrupted/out-of-distribution datasets including ImageNet-A [16] (top-1 accuracy) and ImageNet-C [15] (mCE, lower is better). The generalization ability is evaluated on ImageNet-V2 [36] and ImageNet-Real [2].

| Model | FLOPs | Params | ImageNet | | Generalization | | Robustness | | | |
|---|---|---|---|---|---|---|---|---|---|---|
| | (G) | (M) | Top-1↑ | Top-5↑ | IN-V2↑ | IN-Real↑ | FGSM↑ | PGD↑ | IN-C↓ | IN-A↑ |
| ResNet-50 [14] | 4.1 | 26 | 76.1 | 92.9 | 67.4 | 85.8 | 12.2 | 0.9 | 76.7 | 0.0 |
| ResNeXt50-32x4d [45] | 4.3 | 25 | 79.8 | 94.6 | 68.2 | 85.2 | 34.7 | 13.5 | 64.7 | 10.7 |
| DeiT-S [40] | 4.6 | 22 | 79.8 | 95.0 | 68.4 | 85.6 | 40.7 | 16.7 | 54.6 | 18.9 |
| ResMLP-12 [39] | 3.0 | 15 | 76.6 | 93.2 | 64.4 | 83.3 | 23.9 | 8.5 | 66.0 | 7.1 |
| GFNet-S | 4.5 | 25 | 80.1 | 94.9 | 68.5 | 85.8 | 42.6 | 21.0 | 53.8 | 14.3 |

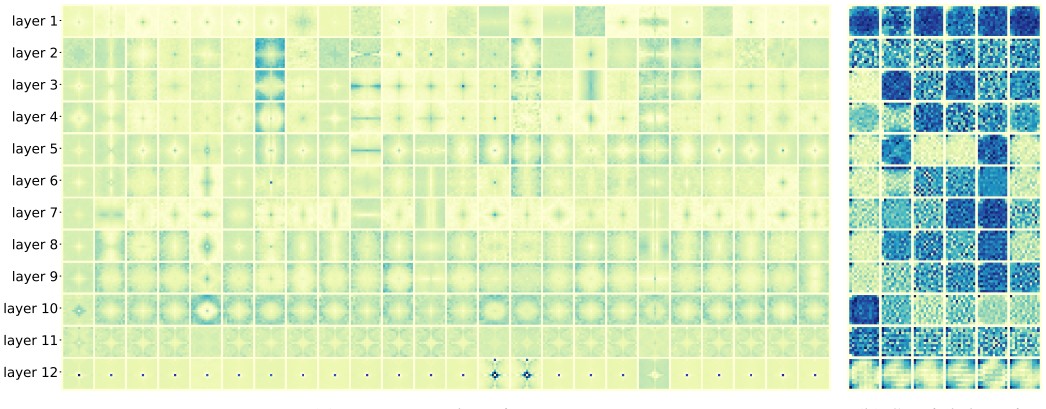

(a) Frequency domain      (b) Spatial domain

Figure 4: Visualization of the learned *global filters* in GFNet-XS. We visualize the original frequency domain global filters in (a) and show the corresponding spatial domain filters for the first 6 columns in (b). There are more clear patterns in the frequency domain than the spatial domain.

**Visualization.** The core operation in GFNet is the element-wise multiplication between frequency-domain features and the global filter. Therefore, it is easy to visualize and interpret. We visualize the frequency domain filters as well as their corresponding spatial domain filters in Figure 4. The learned global filters have more clear patterns in the frequency domain, where different layers have different characteristics. Interestingly, the filters in the last layer particularly focus on the low-frequency component. The corresponding filters in the spatial domain are less interpretable for humans.

## 5 Conclusion

We have presented the Global Filter Network (*GFNet*), which is a conceptually simple yet computationally efficient architecture for image classification. Our model replaces the self-attention sub-layer in vision transformer with 2D FFT/IFFT and a set of learnable *global filters* in the frequency domain. Benefiting from the token mixing operation with log-linear complexity, our architecture is highly efficient. Our experimental results demonstrated that GFNet can be a very competitive alternative to vision transformers, MLP-like models and CNNs in accuracy/complexity trade-offs.

**Acknowledgment**

This work was supported in part by the National Key Research and Development Program of China under Grant 2017YFA0700802, in part by the National Natural Science Foundation of China under Grant 62125603, Grant 61822603, Grant U1813218, Grant U1713214, in part by Beijing Academy of Artificial Intelligence (BAAI), and in part by a grant from the Institute for Guo Qiang, Tsinghua University.

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
