# Global Filter Networks for Image Classification
## Supplementary Material

## A  Discrete Fourier transform

In this section, we will elaborate on the derivation and the properties of the discrete Fourier transform.

### A.1  From Fourier transform to discrete Fourier transform

Discrete Fourier transform (DFT) can be derived in many ways. Here we will introduce the formulation of DFT from the standard Fourier transform (FT), which is originally designed for continuous signals. The FT converts a continuous signal from the time domain to the frequency domain and can be viewed as an extension of the Fourier series. Specifically, the Fourier transform of the signal $x(t)$ is given by

$$X(j\omega) = \int_{-\infty}^{\infty} x(t)e^{-j\omega t}dt := \mathcal{F}[x(t)]. \tag{A.1}$$

The inverse Fourier transform (IFT) has a similar form to the Fourier transform:

$$x(t) = \frac{1}{2\pi}\int_{-\infty}^{\infty} X(j\omega)e^{j\omega t}d\omega. \tag{A.2}$$

From the formulas of the FT and the IFT we can have a glimpse of the duality property of the FT between the time domain and the frequency domain. The duality indicates that the properties in the time domain always have their counterparts in the frequency domain. There are a variety of properties of Fourier transform. To name a few basic ones, the FT of a unit impulse function (a.k.a. Dirac delta function) is

$$\mathcal{F}[\delta(t)] = \int_{-\infty}^{\infty} \delta(t)e^{-j\omega t}dt = \int_{0-}^{0+} \delta(t)dt = 1, \tag{A.3}$$

and the time shifting property:

$$\mathcal{F}[\delta(t-t_0)] = \int_{-\infty}^{\infty} x(t-t_0)e^{-j\omega t}dt = e^{-j\omega t_0}\int_{-\infty}^{\infty} x(t)e^{-j\omega t}dt = e^{-j\omega t_0}X(j\omega). \tag{A.4}$$

However, we rarely deal with continuous signal in the real application. A general practice is to perform *sampling* to the continuous signal to obtain a sequence of discrete signal. The sampling can be achieved using a sequence of unit impulse functions,

$$x_s(t) = x(t)\sum_{n=-\infty}^{\infty} \delta(t-nT_s) = \sum_{n=-\infty}^{\infty} x(nT_s)\delta(t-nT_s), \tag{A.5}$$

where $T_s$ is the sampling interval. Taking the FT of the sampled signal $x_s(t)$ and applying Equation (A.3) and Equation (A.4), we have

$$X_s(j\omega) = \sum_{n=-\infty}^{\infty} x(nT_s)e^{-j\omega nT_s}. \tag{A.6}$$

In the above equation, it is direct to show that $X_s(j\omega)$ is a *periodic* function with the fundamental period as $2\pi/T_s$. Actually, there is always a correspondence between the discrete signal in one

domain and the periodic signal in the other domain. Usually, we prefer a normalized frequency $\omega \leftarrow \omega T_s$ such that the period of $X_s(j\omega)$ is exact $2\pi$. We can further denote $x[n] = x(nT_s)$ as the sequence of discrete signal and derive the discrete-time Fourier transform (DTFT):

$$X(e^{j\omega}) = \sum_{n=-\infty}^{\infty} x[n]e^{-j\omega n}. \tag{A.7}$$

If the discrete signal $x[n]$ has finite length $N$ (which is common in digital signal processing), the DTFT becomes

$$X(e^{j\omega}) = \sum_{n=0}^{N-1} x[n]e^{-j\omega n}, \tag{A.8}$$

where we assume the non-zero terms lie in $[0, N-1]$ without loss of generality. Note that the DTFT is a continuous function of $\omega$ and we can obtain a sequence of $X[k]$ by sampling $X(e^{j\omega})$ at frequencies $\omega_k = 2\pi k/N$:

$$X[k] = X(e^{j\omega})|_{\omega=2\pi k/N} = \sum_{n=0}^{N-1} x[n]e^{-j(2\pi/N)kn}, \tag{A.9}$$

which is exactly the formulation of DFT. The extension from 1D DFT to 2D DFT is straightforward. In fact, The 2D DFT can be viewed as performing 1D DFT on the two dimensions alternatively, *i.e.*, the 2D DFT of $x[m, n]$ is given by:

$$X[u, v] = \sum_{m=0}^{M-1} \sum_{n=0}^{N-1} x[m, n]e^{-j2\pi\left(\frac{um}{M} + \frac{vn}{N}\right)}. \tag{A.10}$$

## A.2 Some properties of DFT

**DFT of real signals.** Given a real signal $x[n]$, the DFT of it is *conjugate symmetric*, which can be proved as follows:

$$X[N-k] = \sum_{n=0}^{N-1} x[n]e^{-j(2\pi/N)(N-k)n} = \sum_{n=0}^{N-1} x[n]e^{j(2\pi/N)kn} = X^*[k]. \tag{A.11}$$

For 2D signals, we have a similar result:

$$\begin{aligned} X[M-u, N-v] &= \sum_{m=0}^{M-1} \sum_{n=0}^{N-1} x[m, n]e^{-j2\pi\left(\frac{(M-u)m}{M} + \frac{(N-v)n}{N}\right)} \\ &= \sum_{m=0}^{M-1} \sum_{n=0}^{N-1} x[m, n]e^{j2\pi\left(\frac{um}{M} + \frac{vn}{N}\right)} = X^*[u, v]. \end{aligned} \tag{A.12}$$

In our GFNet, we leverage this property to reduce the number of learnable parameters and redundant computation.

**The convolution theorem.** One of the most important property of Fourier transform is the convolution theorem. Specifically, for the DFT, the convolutional theorem states that the *multiplication* in the frequency domain is equivalent to the *circular convolution* in the time domain. The circular convolution of a signal $x[n]$ and a filter $h[n]$ can be defined as

$$y[n] = \sum_{m=0}^{N-1} h[m]x[((n-m))_N], \tag{A.13}$$

where we use $((n))_N$ to denote $n$ modulo $N$. Consider the DFT of $y[n]$, we have

$$
\begin{aligned}
Y[k] &= \sum_{n=0}^{N-1} \sum_{m=0}^{N-1} h[m]x[((n-m))_N]e^{-j(2\pi/N)kn} \\
&= \sum_{m=0}^{N-1} h[m]e^{-j(2\pi/N)km} \sum_{n=0}^{N-1} x[((n-m))_N]e^{-j(2\pi/N)k(n-m)} \\
&= H[k]\left( \sum_{n=m}^{N-1} x[n-m]e^{-j(2\pi/N)k(n-m)} + \sum_{n=0}^{m-1} x[n-m+N]e^{-j(2\pi/N)k(n-m)} \right) \quad \text{(A.14)} \\
&= H[k]\left( \sum_{n=0}^{N-m-1} x[n]e^{-j(2\pi/N)kn} + \sum_{n=N-m}^{N-1} x[n]e^{-j(2\pi/N)kn} \right) \\
&= H[k]\sum_{n=0}^{N-1} x[n]e^{-j(2\pi/N)kn} = H[k]X[k],
\end{aligned}
$$

where the right hand is exactly the multiplication of the signal and the filter in the frequency domain. The convolution theorem in 2D scenario can be derived similarly. Therefore, our global filter layer is equivalent to a depth-wise circular convolution, where the filter has the same size as the feature map.

## B    Implementation Details

**The detailed architectures.**    To better compare with previous methods, we use the identical over-all architecture to DeiT Samll [12] and ResMLP-12 [11] for GFNet-XS, where only the self-attention/MLP sub-layers, the final classifier and the residual connection are modified (using a single residual connection in each block will lead to 0.2% top-1 accuracy improvement on ImageNet for GFNet-XS). We set the number of layers and embedding dimension to $\{12, 19, 19\}$ and $\{256, 384, 512\}$ for GFNet-{Ti, S, B}, respectively. The architectures of our hierarchical models are shown in Table 1. We use the similar strategy as ResNet [2] to increase network depth where we fix the number of blocks for the stage 1,2,4 to 3 and adjust the number of blocks in stage 3. For small and base hierarchical models, we adopt the LayerScale normalization [13] for more stable training. The high efficiency of our GFNet makes it possible to *directly* process a large feature map in the early stages (*e.g.*, $H/4 \times W/4$) without introducing any handcraft structures like Swin [7].

**Details about ImageNet experiments.**    We train our models for 300 epochs using the AdamW optimizer [8]. We set the initial learning rate as $\frac{\text{batch size}}{1024} \times 0.001$ and decay the learning rate to $1e^{-5}$ using the cosine schedule. We use a linear warm-up learning rate in the first 5 epochs and apply gradient clipping to stabilize the training process. We set the stochastic depth coefficient [3] to 0, 0, 0.15 and 0.25 for GFNet-Ti, GFNet-XS, GFNet-S and GFNet-B. For hierarchical models, we use the stochastic depth coefficient of 0.1, 0.2, and 0.4 for GFNet-H-Ti, GFNet-H-S, and GFNet-H-B. During finetuning at the higher resolution, we use the hyper-parameters suggested by the implementation of [12] and train the model for 30 epochs with a learning rate of $5e^{-6}$ and set the weight decay to $1e^{-6}$. We set the stochastic depth coefficient to 0.1 for GFNet-S and GFNet-B during finetuning.

**Details about transfer learning experiments.**    We evaluate generality of learned representation of GFNet on a set of commonly used transfer learning benchmark datasets including CIFAR-10 [6], CIFAR-100 [6], Stanford Cars [5] and Flowers-102 [9]. We follow the setting of previous works [10, 1, 12, 11], where the model is initialized by the ImageNet pre-trained weights and finetuned on the new datasets. During finetuning, we use the AdamW optimizer and set the weight decay to $1e^{-4}$. We use batch size 512 and a smaller initial learning rate of 0.0001 with cosine decay. Linear learning rate warm-up in the first 5 epochs and gradient clipping with a max norm of 1 are also applied to stabilize the training. We keep most of the regularization methods unchanged except for removing stochastic depth following [12]. For relatively larger datasets including CIFAR-10 and CIFAR-100, we train the model for 200 epochs. For other datasets, the model is trained for 1000 epoch. Our models are trained and evaluated on commonly used splits following [10]. The detailed splits are provided in Table 2.

Table 1: **The detailed architectures of hierarchical GFNet variants.** We adopt hierarchical architectures where the we use patch embedding layer to perform downsampling. "$\downarrow n$" indicates the stride of the downsampling is $n$. "GFBlock($D$)" represents one building block of GFNet with embedding dimension $D$. We set the MLP expansion ratio to 4 for all the feedforward networks.

| | Output Size | GFNet-H-Ti | GFNet-H-S | GFNet-H-B |
|---|---|---|---|---|
| Stage1 | $\dfrac{H}{4} \times \dfrac{W}{4}$ | Patch Embedding$\downarrow$4 
 GFBlock(64) $\times$ 3 | Patch Embedding$\downarrow$4 
 GFBlock(96) $\times$ 3 | Patch Embedding$\downarrow$4 
 GFBlock(96) $\times$ 3 |
| Stage2 | $\dfrac{H}{8} \times \dfrac{W}{8}$ | Patch Embedding$\downarrow$2 
 GFBlock(128) $\times$ 3 | Patch Embedding$\downarrow$2 
 GFBlock(192) $\times$ 3 | Patch Embedding$\downarrow$2 
 GFBlock(192) $\times$ 3 |
| Stage3 | $\dfrac{H}{16} \times \dfrac{W}{16}$ | Patch Embedding$\downarrow$2 
 GFBlock(256) $\times$ 10 | Patch Embedding$\downarrow$2 
 GFBlock(384) $\times$ 10 | Patch Embedding$\downarrow$2 
 GFBlock(384) $\times$ 27 |
| Stage4 | $\dfrac{H}{32} \times \dfrac{W}{32}$ | Patch Embedding$\downarrow$2 
 GFBlock(512) $\times$ 3 | Patch Embedding$\downarrow$2 
 GFBlock(768) $\times$ 3 | Patch Embedding$\downarrow$2 
 GFBlock(768) $\times$ 3 |
| Classifier | | Global Average Pooling, Linear | | |

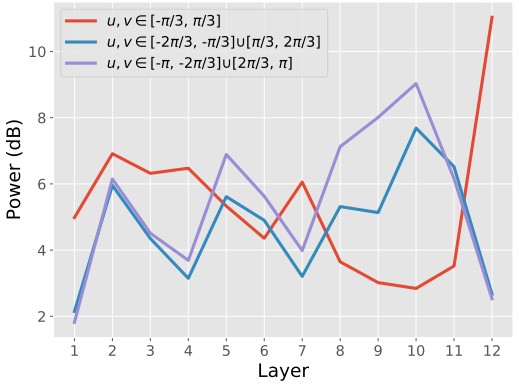

Figure 1: The average power on different frequency ranges of each layer. We can observe that the global filters of different layers focus on different frequencies.

Figure 2: ImageNet accuracy of GFNet and DeiT [12] when directly evaluated on different resolutions without fine-tuning. The GFNet can better adapt to various resolutions.

## C  More Results & Analysis

**Semantic segmentation.** To show the potential of our models on dense prediction tasks, we evaluate our GFNet on ADE20K [15], a challenging semantic segmentation dataset that is commonly used to test vision transformers. We use the Semantic FPN framework [4] and follow the experiment settings in PVT [14]. We train our model for 80K steps with a batch size of 16 on the training set and report the mIoU on the validation set following common practice. We compare the performance and the computational costs of the GFNet series and other commonly used baselines in Table 3. To produce hierarchical feature maps, we adopt the GFNet-H series in the semantic segmentation experiments. We observe that our GFNet works well on the dense prediction task and can achieve very competitive performance in different levels of complexity.

**Power distribution.** We plot the power of the global filters on different frequency ranges of each layer in Figure 1, where we can have a clearer picture of how the global filters of different layers capture the information of different frequencies.

Table 2: **Transfer learning datasets.** We provide the training set size, test set size and the number of categories as references.

| Dataset | Train Size | Test size | #Categories |
|---|---|---|---|
| CIFAR-10 [6] | 50,000 | 10,000 | 10 |
| CIFAR-100 [6] | 50,000 | 10,000 | 100 |
| Stanford Cars [5] | 8,144 | 8,041 | 196 |
| Flowers-102 [9] | 2,040 | 6,149 | 102 |

Table 3: **Semantic segmentation results on ADE20K.** We report the mIoU on the validation set. All models are equipped with Semantic FPN [4] and trained for 80K iterations following [14]. The FLOPs are tested with $1024 \times 1024$ input. We compare the models that have similar computational costs and divide the models into three groups: 1) tiny models using ResNet-18, PVT-Ti and GFNet-H-Ti; 2) small models using ResNet-50, PVT-S, Swin-Ti and GFNet-H-S and 3) base models using ResNet-101, PVT-M, Swin-S and GFNet-H-B.

| Backbone | *Tiny* | | | *Small* | | | *Base* | | |
|---|---|---|---|---|---|---|---|---|---|
| | FLOPs | Params | mIoU | FLOPs | Params | mIoU | FLOPs | Params | mIoU |
| ResNet [2] | 127 | 15.5 | 32.9 | 183 | 28.5 | 36.7 | 260 | 47.5 | 38.8 |
| PVT [14] | 123 | 17.0 | 35.7 | 161 | 28.2 | 39.8 | 219 | 48.0 | 41.6 |
| Swin [7] | - | - | - | 182 | 31.9 | 41.5 | 274 | 53.2 | 45.2 |
| GFNet-H | 126 | 26.6 | 41.0 | 179 | 47.5 | 42.5 | 261 | 74.7 | 44.8 |

**Directly adapting to other resolutions.** As is discussed in Section **??**, one of the advantages of GFNet is the ability to deal with arbitrary resolutions. To verify this, we *directly* evaluate GFNet-S trained with $224 \times 224$ images on different resolutions (from 128 to 448). We plot the accuracy of GFNet-S and DeiT-S in Figure 2 and find our GFNet can adapt to different resolutions with less performance drop than DeiT-S.

**Filter visualization for hierarchical models.** We also provide the visualization of the frequency-domain global filters for the hierarchical model GFNet-H-B in Figure 3.