# OpenReview forum: "Global Filter Networks for Image Classification"
_NeurIPS.cc/2021/Conference — NeurIPS 2021 Poster_

### Official Review · Reviewer_niRJ · 2021-07-13

**Rating:** 6
**Confidence:** 4

**Summary:**

The work proposes a new attention-based architecture for computer vision (classification). The proposed architecture replaces the self-attention layer in ViT with computational blocks involving the Fourier transform. Due to the efficiency of FFT and IFFT the quadratic complexity of self-attention can be reduced to a log-linear computational cost. In order to boost performance distillation of taken features from a pre-trained vision transformer is used. Results are presented on ImageNet as well as on some transfer learning experiments.

**Limitations And Societal Impact:**

I am not particularly convinced by the motivation of "removing inductive biases" -- why? Certainly with modern accelerators and advanced training tricks we can match or sometimes exceed the performance of ConvNets, but it seems to me that the inductive biases embodied by ConvNets are still relevant and useful. Not only that, but these inductive biases are often based on domain knowledge and give insight to the problem they solve.

**Main Review:**

Conceptually, the proposed model seems to be very close to a ConvNet with large filters. L145 states that the global filter layer is equivalent to a circular convolution with a large filter. Furthermore because these layers are stacked (L195-196), it because to look very close to a ConvNet. Furthermore, a very similar model (FNet, 18) was recently proposed for NLP. The motivation of this work is to remove inductive biases from modern architectures (largely based on ConvNets), however it seems to me that in attempting to do so we arrived back at stacked convolutions (computed in the frequency domain). This model doesn't seem to offer a great performance advantage, considering that the SoTA top-1 performance on ImageNet is >90% (large vision transformer). For example, without distillation, GFNet-12 achieves 78.5% while EfficientNet-B3 achieves 81.6% with fewer trainable parameters.

**Time Spent Reviewing:**

1.5

---

> ### Author Response · Authors · 2021-08-10
> **Response to Reviewer niRJ**
>
> Thanks for your comments. We believe our main contribution in this paper is providing a new basic building block for vision understanding. We agree that ConvNet is a very successful framework in computer vision, but we want to emphasize that exploring new frameworks is also valuable for the community. GFNet has several distinct contributions compared to previous works:
>
> 1) **Previous ConvNets usually consist of small and local convolutions, while we use global filters to learn visual representations.** To our best knowledge, we present the first attempt that uses only global convolutions and MLP to achieve competitive performance on ImageNet. Our solution based on FFT operations is much more efficient than vanilla large kernel convolutions thanks to the log-linear complexity. Our visualization presented in Figure 4 also shows the learned global filters have more clear patterns in the frequency domain than the spatial domain, which is completely different from conventional local convolutions.   Besides, our method is simpler since we don’t need to tune hyper-parameters like kernel sizes, padding strategy, etc. Although our model may not outperform some carefully designed ConvNets like EfficientNet (which requires quite a few GPU resources to perform architecture search), we provide a new and practical idea for designing efficient vision models. We believe the proposed global filter layer provides a new element in the network design space.  Combining the proposed method with existing self-attention and convolution modules can be a promising way to design state-of-the-art vision models.
>
> 2) **GFNet is distinct from FNet in many aspects.** Although both works use FFT to efficiently mix tokens, we present a more effective architecture for vision problems. As shown in Table 4, GFNet can outperform FNet by 7.3% on ImageNet with less computation. Besides, FNet is a concurrent work of GFNet. FNet was released on arXiv in May 2021 while our work was also submitted to NeurIPS in May. We also listed three key differences in Line 183-191.
>
> 3) **Removing inductive biases can reduce hand-made choices and help the models to better learn from data.** We don’t need to tune hyper-parameters like kernel sizes for our model.  Since we consider a global filter in our model, a local convolution can be viewed as a subset/part of the global filter. We agree that the inductive biases can give insight into the problem they solve, but our method on the contrary provides a more general and flexible way to learn from data. Recent progress in large vision transformers also proves that vision models can achieve state-of-the-art performance with less inductive biases [a].
>
> 4) **GFNet also shows some unique advantages in efficiency, generalization ability and robustness.**  While EfficientNet-B3 can achieve better accuracy on ImageNet, GFNet-12 is nearly 2 times faster on GPU (1811 images/s vs. 982 images/s). As shown in Table 3,  GFNet also shows comparable performance with EfficientNet-B7 on transfer learning tasks with only 8% FLOPs. In Appendix C, we also show that GFNet can achieve even better trade-offs on semantic segmentation where large feature maps are required. GFNet-H18 with higher throughput and lower FLOPs/#Param can achieve the same mean IoU with the state-of-the-art Swin Transformer. Our recent experiments also show GFNet is more robust to the adversarial attack compared to CNNs and ViT. Specifically, under PGD attack, 19-layer GFNet can achieve 21.0% accuracy while ResNeXt50-32x4d and DeiT-S with similar FLOPs can obtain 13.5% and 16.7% accuracy.  We will add more discussion and results in the revised paper.
>
> More details about our experiments on network robustness can be found in the following table. GFNet-19 is a deeper version of GFNet-12, which has a similar complexity with ResNet-50 and DeiT-S while having better performance.
>
> | Model                                    | GFLOPS | Params (M) | ImageNet Top-1 $\uparrow$ | FGSM [b] $\uparrow$ | PGD [c] $\uparrow$ | ImageNet-C [d]  $\downarrow$ | ImageNet-V2 [e] $\uparrow$ |
> |---------------------------------|-------|--------|----------------|--------|---------|---------|------|
> | ResNet-50            | 4.1   | 26     | 76.1           | 12.2 | 0.9  | 76.7 | 67.4  |
> | ResNeXt50-32x4d | 4.3   | 25     | 79.8           | 34.7 | 13.5 | 64.7 | 68.2 |
> | DeiT-S            | 4.6   | 22     | 79.8           | 40.7 | 16.7 | 54.6 | 68.4 |
> | GFNet-19                                  | 4.5   | 25     | 80.1           | 42.6 | 21.0 | 53.8 | 68.5 |
>
>
> We hope our response could address the concerns, and we thank you again for the helpful comments. We are glad to discuss further comments and suggestions.
>
>
> [a] Scaling Vision Transformers, arXiv
>
> [b] Explaining and harnessing adversarial examples, ICLR 2015
>
> [c] Towards deep learning models resistant to adversarial attacks, ICLR 2018
>
> [d] Benchmarking neural network robustness to common corruptions and perturbations, ICLR 2019
>
> [e] Do Imagenet classifiers generalize to Imagenet? ICML 2019

---

> > ### Comment · Reviewer_niRJ · 2021-08-17
> > **Upgrading my score**
> >
> > Thank you for your clarifications, in light of the reasons stated below I have updated my review score.
> >
> > Reasons for revised score:
> > 1-I did not realize this work was concurrent with FNet.
> > 2-I did not fully appreciate the simplicity and computational speed up afforded by the proposed model.
> >
> > Reasons why the score is not even higher:
> > Because it is not clear what the compute/performance tradeoff should be, I consider works that strictly improve on both to be clear improvements. The present work requires a sacrifice in performance for a computational speedup.  e.g. "While EfficientNet-B3 can achieve better accuracy on ImageNet, GFNet-12 is nearly 2 times faster on GPU..."

---

> > > ### Author Response · Authors · 2021-08-19
> > > **Thanks for upgrading the score and providing valuable feedback**
> > >
> > > Thanks for upgrading the score and providing valuable feedback. We would like to further discuss your concerns on the complexity/accuracy trade-off of our GFNet. Recently, we continue to explore the possibility of using our simple building block to design more powerful models. We found the performance of the GFNet-H series can be further improved by optimizing the number of blocks in each stage and the training strategies. Specifically, we obtained 3 new models by adjusting the number of blocks/channels for the four stages and setting the gradient norm to 1 to stabilize training.  We kept the other details in Appendix B unchanged and didn’t use the token-wise distillation technique for fair comparisons. The detailed architectures of the models and the results are described in the following table (we use $x_1$-$x_2$-$x_3$-$x_4$ to show the number of blocks/channels of the four stages).
> > >
> > > | Name       | #Blocks  | #Channels      | #Params (M) | GFLOPs | ImageNet Top-1 (%)|  ImageNet Top-5 (%) |
> > > |------------|----------|----------------|---------|--------|-------|-------|
> > > | GFNet-H-Ti | 3-3-10-3 | 64-128-256-512 | 15      | 2.0    | 80.1  | 95.1  |
> > > | GFNet-H-S  | 3-3-10-3 | 96-192-384-768 | 32      | 4.5    | 81.5  | 95.6  |
> > > | GFNet-H-B  | 3-3-27-3 | 96-192-384-768 | 54      | 8.4    | 82.9  | 96.2  |
> > >
> > > We test the speed (measured by GPU throughput and CPU latency) of the above models and compare them with the EfficientNet series. The results are summarized as below:
> > >
> > > |                 | GPU throughput (images / s) | CPU latency (ms) | ImageNet Top1 (%) |
> > > |-----------------|-----------------------------|------------------|-------------------|
> > > | EfficientNet B2 | 1654                        | 251              | 80.1              |
> > > | GFNet-H-Ti      | 1744 **(+5.5%)**                | 63 **(-74.7%)**      | 80.1 (+0.0)       |
> > > | EfficientNet B3 | 982                         | 362              | 81.6              |
> > > | GFNet-H-S       | 1173 **(+19.4%)**               | 113 **(-68.7%)**     | 81.5 (-0.1)       |
> > > | EfficientNet B4 | 483                         | 593              | 82.9              |
> > > | GFNet-H-B       | 718 **(+48.6%)**                | 201 **(-66.1%)**      | 82.9 (+0.0)       |
> > >
> > > We find GFNet-H series run faster than their EfficientNet counterparts with similar accuracy on both GPU and CPU (more significantly). These results show GFNet with a simple 4-stage architecture can achieve better trade-offs than EfficientNet.  We believe models based on our global filter layer can achieve even better performance by combining local convolutions and self-attention layers or using NAS methods to carefully tune architectures.  We hope the above analysis can address your concerns.

---

### Official Review · Reviewer_6RZZ · 2021-07-14

**Rating:** 7
**Confidence:** 5

**Summary:**

This paper presents global filter network, a simple and efficient architecture for visual recognition. Unlike transformer-based visual recognition models that utilize self-attention to encode spatial information, this paper proposes to build long-range dependencies among spatial locations in the frequency domain by taking advantage of 2D FFT and IFFT.

**Limitations And Societal Impact:**

- In the experiment section, the authors say that strong data augmentation methods are used for training. However, these are no experiments demonstrating how each data augmentation method contributes to the model performance. I think an ablation path on data augmentation should be given.

- The authors use token-wise distillation strategy while training. Have you attempted to use the original token labeling while training? Does the token-wise distillation strategy perform better than token labeling?

- It is great to see the visualization of the learned filters. It would be better if the authors could provide more explanations on what the proposed approach learns in terms of the filters.

**Main Review:**

- The originality of this paper is clear. Encoding spatial information in the frequency domain is interesting. The classification performance on ImageNet is also good.

- This paper is also well presented. It is simple and easy to follow.

**Time Spent Reviewing:**

3

---

> ### Author Response · Authors · 2021-08-10
> **Response to Reviewer 6RZZ**
>
> We sincerely thank the reviewer for the positive comments on our work! We address the questions and clarify the issues accordingly as described below.
>
> **Q1: About the strong data augmentation methods**
>
> **[Reply]** Thanks for your suggestion. To fairly compare to previous works, we directly adopt the data augmentation methods used in DeiT and Swin Transformer without any modifications for GFNet and GFNet-H series, respectively. As these strong data augmentation methods are commonly used in recent works on vision transformers, we didn’t conduct ablation studies on these methods. Since it is difficult to finish all the ablation studies in a week, we will conduct these experiments later and include the results in the revised paper.
>
> **Q2: Token-wise Distillation**
>
> **[Reply]** We have also tried the original Token Labeling strategy. However, the GFNet-12 trained using the Token Labeling codebase can only obtain 78.1% top-1 accuracy on ImageNet, which is 0.4% lower than GFNet-12 without token-wise distillation and 1.6% lower than GFNet-12 with token distillation. We find that MixUp augmentation is critical to train GFNet, while the original Token Labeling is not compatible with MixUp. Since mixing up images in the spatial domain is equivalent to mixing up the spectrums of images, we think MixUp is a very effective way to improve the generalization ability of models based on frequency-domain learning. Compared to the original Token Labeling, the proposed token-wise distillation can simultaneously overcome the over-smoothing issue while applying MixUp augmentation during training, which is more suitable for our GFNet models.
>
> **Q3: Visualization**
>
> **[Reply]** Thanks for your insightful advice. We found filters from different layers tend to capture different frequency-domain patterns. In general, the shallow layers focus more on low-frequency patterns while deeper layers (except the last one) tend to learn high-frequency patterns. The filters in the last layer are mostly low-pass filters. We will add more analysis and visualization in the revised paper.

---

> > ### Comment · Reviewer_6RZZ · 2021-08-11
> > **Response to the rebuttal**
> >
> > Thanks for the responses from the authors. The idea of this paper is interesting and the authors have solved most of my concerns. I tend to keep my original rating unchanged.

---

### Official Review · Reviewer_yCXs · 2021-07-15

**Rating:** 7
**Confidence:** 3

**Summary:**

This paper proposes to replace self-attention in vision transformer by a *Global Filter Layer* consisting of a 2D fast Fourier transform (FFT), a point-wise multiplication with learned weights and an 2D inverse FFT. The proposed layer compares favourably with (a) self-attention as it replaces the quadratic dependency on compute cost and memory footprint with a log-linear rate and (b) variants of MLP-mixer as the number of weights scales linearly with the number of pixels instead of quadratically. The performances of the proposed model are convincing both for supervised classification on ImageNet and fine-tuning to downstream task on smaller datasets.

**Main Review:**


I would like to thank the authors for this very interesting piece of work. This is a timely contribution to the field and a very interesting read.

The introduced Global Filter Layer is simple and well motivated. The idea is novel and allows to avoid the quadratic dependency in attention models while reaching similar performance. It also solves two problems inherent to MLP-Mixer: (1) it requires only a linear number of parameters in term of input size and (2) moving to the Fourier domain gives a principled approach to upscale a trained model to process images at a different resolution.
I find the contribution solid and I would tend to accept the paper.

### Questions and concerns

I am not sure what is the common way to account for complex multiplications but if we consider it requires 4 FLOPS, should there be a factor 2 in the second term of the number of FLOPS in Table 1? (accounting that you halve the size of the signal with line 161).

Concerning Figure 4 (a) frequency domain, can you clarify if you are plotting the module of the weights or their real part maybe?

I found in the code that you use a `num_heads` parameter in `GlobalFilter` (file `code/gfnet.py` line 58) which is not mentioned in the paper. Could you clarify its usage? If I understand the code correctly, it seems that some weights in $\mathbf{K}$ (eq. 3.5) are shared.
This decreases the number of parameters of the layers by a factor $D / N_{heads}$ but constraints the diversity of frequency filters that can be learned. Could you comment if the model proposed in the paper ($N_{heads} = D$) performs well in practice or if you would advise for your alternative with heads?

Beyond proposing a novel *Global Filter Layer*, the authors further push the performance of their model with some interesting techniques that are often orthogonal to the main contribution:

* The best performing models are the hierarchical version H18 and H24 and they are only introduced in the Appendix. Please detail how you chose the hyper-parameters for this architecture. Even though the performance is great and the architecture interesting, this goes against the general idea of MLP-mixers to introduce ``fewer inductive biases'' and requires more hyper-parameter tuning.

* Token-wise distillation could be applied to other baselines to be fair (except ConvNets). Same goes for upscaling to 384 for vision transformers (which is standard) but not for MLP-like model where the proposed model introduces a principled approach for upscaling.


### Checklist

3. (d). Please provide the total amount of compute needed to train your model on your 8 GPUs machine. Can you also provide in the Appendix a version of Figure 3 with the training time as x-axis?


**Time Spent Reviewing:**

6

---

> ### Author Response · Authors · 2021-08-10
> **Response to Reviewer yCXs**
>
> We sincerely thank the reviewer for the positive comments on our work! We address the questions and clarify the issues accordingly as described below.
>
>
> **Q1: About the FLOPs**
>
> **[Reply]** Sorry for the mistakes. The second term should be $4\times H\times \frac{W}{2}\times D=2HWD$ and we have missed a factor 2 here. We will fix this in the revised paper. It is worth noting that this term only contributes <0.1GFLOPs  in the whole model. Thus, the trade-off shown in the experiment part will not be significantly affected.
>
> **Q2: About the visualization of Figure 4 (a)**
>
> **[Reply]** Sorry for the confusion. We use the modulus of the weights to plot the visualization in Figure 4. The modulus of the weights can represent how the global filter layer changes the magnitude of the spectrum. We will clarify this in the revised version.
>
> **Q3: About ```num_heads```**
>
> **[Reply]** Thanks for the interest in our work. The ```num_heads```is a legacy parameter and we have found setting it to another value instead of $D$ will be harmful to the performance. Therefore, in all the experiments, we do not use this parameter. We will remove this parameter in the finally released code.
>
> **Q4: About the hierarchical variants of GFNet**
>
> **[Reply]** When designing the hierarchical variants of GFNet (GFNet-H-18 and GFNet-H-24), we mainly follow the principle in ResNet and Swin Transformer. Specifically, we use a 4-stage architecture, where stage 1, 2, 4 have fewer blocks and stage 3 has relatively more blocks. We mainly adjust the number of blocks in stage 3 and the channel dimension to scale the model. We agree that hierarchical variants of GFNet will bring more hyper-parameters. By introducing these hierarchical variants, we mainly want to show the possibilities of further improving the performance with only the global filter layers and MLPs. GFNet is simpler while GFNet-H has better performance.
>
> **Q5: About token-wise distillation and upscaling to 384**
>
> **[Reply]** When we compare the GFNet to the baseline models, we always use the vanilla version. The experiments with token-wise distillation just aim to show the performance of GFNet can be further improved with some extra techniques. Similarly, the experiments with upscaling to 384 are used to prove the scalability of GFNet to higher resolutions. We also note that the vanilla GFNet series are already competitive to the baseline methods: GFNet-12 outperforms ResMLP-12 by large margin and GFNet-19 (with 19 blocks) exhibit better accuracy/complexity trade-off (80.0% accuracy on ImageNet with 4.5 GFLOPs) than DeiT (79.8% accuracy with 4.6 GFLOPs). We will add more discussions and results in the revised paper.
>
> **Q6: Training time**
>
> **[Reply]** Thanks for the valuable advice. It takes 34.6h to train a GFNet-12 model on our 8 RTX2080Ti GPUs machine. For reference, the total training time for DeiT is 40.3h. We will provide a figure similar to Figure 3 with the total training time as x-axis in the revised paper.

---

> > ### Comment · Reviewer_yCXs · 2021-08-17
> > **Reviewer yCXs's answer**
> >
> > I would like to thank the authors for their answer. They have answered all my questions. I look forward to seeing dissemination of this work and future applications.

---

### Official Review · Reviewer_wrgX · 2021-07-16

**Rating:** 7
**Confidence:** 4

**Summary:**

This paper proposes Global Filter Networks (GFNet), which make use of discrete Fourier transform (DFT) and a global filter layer to mix the input tokens. According to the convolution theorem, the combination of the two operations can be regarded as a depthwise global circular convolution, and thus this work can be viewed as a kind of convolutional network with global convolutional layers. By taking advantage of FFT, the proposed method is faster than previous methods such as self-attention and spatial MLP with comparable performance on ImageNet.

**Limitations And Societal Impact:**

The authors said the limitations can be found in Section 5, but I did not find them. The authors should include them.


**Main Review:**

Pros:
+ The work is well-written and easy to understand.
+ According to the experiments, the proposed method is advantageous. Compared to Vit and ResMLP, the method has less latency and memory footprint, as shown in Fig. 2. It also achieves comparable or better image classification performance than these works, as shown in Table 4.
+ Nice tricks are used to speed up the computation by using the conjugate symmetry of DFT.

Cons:
- The novelty is limited. It is well-known that in the frequency domain, the convolution operation can be transformed into an element-wise multiplication according to the convolution theorem. The paper can be viewed as an application of the fact, but the fact has actually been widely used in CNNs.
- The proposed method does not have obvious advantages over traditional CNNs. As shown in Table 4, compared with "Local Conv (7x7)", GFNet-12 achieves 78.5 vs 78.2 with similar FLOPS. Given the well-optimized implementation of CNNs, I am pretty sure that the actual latency of GFNet will be longer than CNNs, which makes GFNet less practical.


**Time Spent Reviewing:**

2 hours

---

> ### Author Response · Authors · 2021-08-10
> **Response to Reviewer wrgX**
>
> We sincerely thank the reviewer for the positive comments on our work! We address the questions and clarify the issues accordingly as described below.
>
>
> **Q1: About the novelty**
>
> **[Reply]** This work follows the line of work of designing transformer-like and MLP-like architecture. We aim to seek an efficient and effective way to interchange information among different tokens. Although the convolution theorem is well known, we are the first to leverage the FFT to build the Global Filter Network with only the global filters and MLPs (global filter is rarely seen in CNN architectures). Therefore, our main contribution lies in the efficient architecture that enjoys favorable accuracy/complexity trade-offs and several nice properties.
>
> **Q2: About the speed**
>
> **[Reply]** We provide a detailed analysis of the wall-clock time of GFNet and some variants in the following table. We use a single NVIDIA RTX 3090 GPU with batch size 32 to test the GPU throughput. We find our GFNet has slightly longer latency than $7\times 7$ local convolution on GPU but is faster than local convolutions with larger kernel sizes. The reason is that the convolution operations with small kernel sizes are highly optimized by modern libraries like CuDNN. We also find our GFNet is surprisingly faster than the convolution with any kernel sizes on CPU (the results are tested on an Intel i9-10900X CPU using PyTorch). The performance of our model can be improved if FFT operators are further optimized on GPU in the future.
>
> | model | GPU throughput (images/s)| CPU latency (ms)|
> |----|-----|-----|
> |DeiT-S | 1415.7 |24.8|
> |GFNet-12| 1811.8 | 20.4 |
> | Local Conv ($7\times 7$) | 2165.1 | 120.5 |
> | Local Conv ($9\times 9$)  | 1302.9 | 144.9 |
> | Local Conv ($11\times 11$) | 1114.5 | 172.4 |
> | Local Conv ($13\times 13$)  | 958.4 | 208.3 |
>
> **Q3: Limitations and societal impact**
>
> **[Reply]** Thanks for the suggestion. We will add more discussions in the revised paper.

---

> > ### Comment · Reviewer_wrgX · 2021-09-03
> > **After-rebuttal**
> >
> > I have read the rebuttal and decided to retain my score.

---

### Official Review · Reviewer_oVSg · 2021-07-26

**Rating:** 7
**Confidence:** 4

**Summary:**

This paper replaces the self-attention layer with a global filter layer:  a sequence of a 2D FFT, einsum, and then 2D inverse FFT. A benefit of the approach is that the attention computation is reduced from O(n^2) to O(nlogn) time-complexity. This demonstrates that GFNet is Pareto optimal versus other Transformer-like models on ImageNet on an accuracy-FLOPs basis.

**Limitations And Societal Impact:**

Yes.

**Main Review:**

I am positively inclined towards this paper. Their approach appears to improve over other concurrent efforts to use Fourier Transforms like FNet (Lee-Thorp, 2021).

Questions:
* Putting aside FLOPs, how fast are these on modern accelerators (e.g GPUs, TPUs)? For instance, certain operations, like depthwise convolutions, while FLOP-efficient, are relatively slow on a wall-clock basis. Slow performance doesn’t disqualify the idea of using FFT, but it should be noted here.
* For many larger models, it seems that the O(n^2) time-complexity of self-attention operations are usually not the bottleneck. Could the authors comment on how this improvement scales with model size?
* Does the visualization of the learned features in the frequency domain suggest further improvements to the model? Is diversity of features important to model performance?
* Did the authors consider the high resolution applications discussed in the paper? Given the scalability benefit of the technique, a positive result here might increase the impact of this work.
* Can the paper dedicate more details to the token-wise distillation in the main body? Given the +1% boost to performance, it might be useful to present more details to the reader earlier.


**Time Spent Reviewing:**

5 hrs

---

> ### Author Response · Authors · 2021-08-10
> **Response to Reviewer oVSg**
>
> We sincerely thank the reviewer for the positive comments on our work! We address the questions and clarify the issues accordingly as described below.
>
> **Q1: About the wall-clock time**
>
> **[Reply]** We test the GPU throughput on a single NVIDIA RTX 3090 GPU with batch size 32. We also measure the CPU latency on an Intel i9-10900X CPU and set the batch size to 1. We find the GFNet-12 is significantly faster than DeiT-S. We also test some variants by replacing the global filter layer with the local convolutions. GFNet-12 is slightly slower than $7\times 7$ Conv but faster than $9\times 9\sim 13\times 13$ Conv. GFNet-12 is also much faster than local convolution on the CPU. These results suggest that convolution with small kernel sizes ($3\times 3$, $\times $, $7\times 7$) are highly optimized by modern architectures and acceleration libraries like CuDNN. The performance of our model can be improved if FFT operators are further optimized on GPU in the future.
>
> | model | GPU throughput (images/s)| CPU latency (ms)|
> |----|-----|-----|
> |DeiT-S | 1415.7 |24.8|
> |GFNet-12| 1811.8 | 20.4 |
> | Local Conv ($7\times 7$) | 2165.1 | 120.5 |
> | Local Conv ($9\times 9$)  | 1302.9 | 144.9 |
> | Local Conv ($11\times 11$) | 1114.5 | 172.4 |
> | Local Conv ($13\times 13$)  | 958.4 | 208.3 |
>
> **Q2: How does the improvement on complexity scale with model size?**
>
> **[Reply]** Our GFNet can reduce the computational complexity by (1) replacing the heavy self-attention by the global filter layer; (2) saving costs from the 4 fully connected (FC) layers in the attention block (the three FC layers to compute q, k, v, and the projection layer to produce the output). Therefore, the GFNet can always reduce more computational costs as the model size increases in any of the following cases: (1) using higher resolutions; (2) increasing the number of channels; (3) stacking more blocks. We also find in the experiments that a GFNet-19 (a deeper version of GFNet-12 that consists of 19 blocks) has lower FLOPs (4.5G) than DeiT (4.6G) but can achieve better accuracy on ImageNet (80.0%) than DeiT (79.8%).
>
> **Q3: About the visualization**
>
> **[Reply]** Actually, we just find it would be interesting to visualize what the GFNet learns in each layer in the frequency domain. Our main finding is that the GFNet tends to learn more clear patterns in the frequency domain instead of the spatial domain. Although there may not be a clear relationship between the diversity of the learned filters and the final performance, we believe that diverse filters might indicate that the model captures less redundant information and thus is more efficient.
>
> **Q4: About the high-resolution applications**
>
> **[Reply]** The scalability of GFNet enables us to perform tasks like segmentation where the input resolution is larger. In the Supplementary (Appendix C), we have shown that the GFNet can achieve comparable performance with the state-of-the-art Swin Transformer on ADE20K, where high-resolution features are required to predict per-pixel semantic labels. We will consider moving the segmentation results to the main body of our revised paper.
>
> **Q5: About the token-wise distillation**
>
> **[Reply]** Thanks for the constructive advice. The token-wise distillation is used to overcome over-smoothing and the details have been elaborated in the Supplementary (Appendix B). We will describe this technique in the main body of the revised paper to make it more clear.

---

### Decision · Program_Chairs · 2021-09-27

**Decision:**

Accept (Poster)

**Comment:**

This work proposes replacing self-attention with a global filter layer and applies this method to image classification problems. The resulting method replaces O(n^2) attention operation with a O(nlogn) time-complexity while achieving favorable performance in computation time vs accuracy trade-off curves. The reviewers commented positively on the motivation, clarity of exposition, and the implementation. Although the reviewers noted some concerns about novelty with respect to previous architectures, overall the reviewers were favorable to the acceptance of this work. Despite these minor concerns, this paper is accepted to the conference.